# ROYAL SOCIETY
# OPEN SCIENCE

Subject Areas:
evolution/ecology

Keywords:
time-calibrated phylogeny, phylogenetic regression, Köppen–Geiger climate class, climate niche, crop wild relatives

# Interactions between breeding system and ploidy affect niche breadth in *Solanum*

Nathan Fumia[1], Daniel Rubinoff[2], Rosana Zenil-Ferguson[3], Colin K. Khoury[4,5], Samuel Pironon[6], Michael A. Gore[7] and Michael B. Kantar[1]

[1]Department of Tropical Plant and Soil Science,[2]Department of Plant and Environmental Sciences, and [3]School of Life Sciences, University of Hawaii at Manoa, Honolulu, HI, USA
[4]International Center for Tropical Agriculture (CIAT), Cali, Colombia
[5]San Diego Botanic Garden, Encinitas, CA, USA
[6]Royal Botanic Gardens, Kew, Richmond, UK
[7]Plant Breeding and Genetics Section, School of Integrative Plant Science, Cornell University, Ithaca, NY 14853, USA

NF, 0000-0003-1933-1902; DR, 0000-0002-2732-3032;
RZ-F, 0000-0002-9083-2972; CKK, 0000-0001-7893-5744;
SP, 0000-0002-8937-7626; MAG, 0000-0001-6896-8024;
MBK, 0000-0001-5542-0975

Author for correspondence:
Michael B. Kantar
e-mail: mbkantar@hawaii.edu

Understanding the factors driving ecological and evolutionary interactions of economically important plant species is important for agricultural sustainability. The geography of crop wild relatives, including wild potatoes (*Solanum* section *Petota*), have received attention; however, such information has not been analysed in combination with phylogenetic histories, genomic composition and reproductive systems to identify potential species for use in breeding for abiotic stress tolerance. We used a combination of ordinary least-squares (OLS) and phylogenetic generalized least-squares (PGLM) analyses to identify the discrete climate classes that make up the climate niche that wild potato species inhabit in the context of breeding system and ploidy. Self-incompatible diploid or self-compatible polyploid species significantly increase the number of discrete climate classes within a climate niche inhabited. This result was sustained when correcting for phylogenetic non-independence in the linear model. Our results support the idea that specific breeding system and ploidy combinations increase niche breadth through the decoupling of geographical range and niche diversity, and therefore, these species may be of particular interest for crop adaptation to a changing climate.

# 1. Introduction

Potato (*Solanum tuberosum* L.) is the most important tuber crop worldwide and is the fourth most important crop globally [1]. There is a lack of genetic diversity among many crops, including *S. tuberosum* [2,3], placing increased pressure upon crop management protocols to maintain food security. A proven approach to increasing genetic diversity in crop species is through the utilization of wild relatives for crop improvement [2,4]. Cultivated potato has 199 known wild relatives, forming the *Solanum* section *Petota*, inhabiting 16 countries in the Americas, and ranging from 38° N to 41° S [5]; 72 of the most threatened and useful species to humans have recently been prioritized for conservation [1]. These 72 species are most found in tropical highlands at 600–1200 m in elevation and possess phenotypes similar to cultivated potato through the production of a starchy tuber [6].

Given the importance of maintaining crop productivity, many attributes of the wild relatives of *S. tuberosum* have been defined; their ploidy, breeding system, germplasm classification, endosperm balance number, phylogenies and geographical ranges [1,5,7–9]. These data can be used to discover novel beneficial characteristics present within the wild relative germplasm. Furthermore, research has identified potato as one of the crops in sub-Saharan Africa with the highest potential to benefit from crop wild relatives (CWRs) for climate change adaptation; however, these results have not been integrated with biological (e.g. breeding system and ploidy) and evolutionary (e.g. phylogenetic tree) information [10]. Despite the wide array of information surrounding the wild relatives of potato, one attribute continues to be under-defined—the discrete climate zones each species inhabits that in aggregate make up the species niche and the factors involved (e.g. breeding system, ploidy) in driving the evolution of the highly dynamic climatic diversity in *Solanum* section *Petota*.

Individually exploring life-history traits [11–14] such as the breeding system has led to contradictory conclusions regarding these traits' influence on ecological range [9,15–20], while other traits such as ploidy [21] have shown a consistent influence. For example, diversification models Zenil-Ferguson *et al*. [22] showed that ploidy is the most probable pathway to evolve self-compatibility across Solanaceae. Therefore, there exists an important interaction between ploidy and breeding system [9,16,23] that might impact evolutionary and ecological processes [14]. Polyploidization facilitates self-compatibility as whole genome duplication provides security against inbreeding depression [9,16,22,23]; whereas, self-compatible diploid populations often suffer from large inbreeding depression [24,25]. As a result, diploid populations are more reliant on self-incompatibility to drive adaptive evolution. In Solanaceae, polyploid species show higher rates of self-compatibility [9,16,23]. This clear interaction between ploidy and breeding systems provides the opportunity to test a key hypothesis: that self-compatible species rely on polyploidy in order to generate the variation they need to colonize wider niche space (i.e. more discrete climate zones).

To identify the driving factors of ecological diversity in potato wild relatives, we investigated two key biological aspects of ecological diversity—breeding system and ploidy in 72 wild relatives of potato. We combined species' occurrence, climatic, biological (e.g. breeding system and ploidy) and phylogenetic tree of *Solanum* taxa to test whether the niche breadth (i.e. number of discrete climates that make up the climate niche) of a given species is guided by a specific breeding system and ploidy interaction. To account for the potential decoupling of geographical range and niche breadth [26], the measure of climatic diversity is through the use of discrete climate-classification of each occurrence of these wild relative species. This approach has the added benefit of providing operationalizable information for plant breeders by identifying the optimal climate to use a CWR for plant breeding. This work supports classic ecological theory of niche divergence without the requirement of inferring continuous species distributions from point-based climate descriptions by featuring the relationship between two common intrinsic factors of niche expansion: (i) decreased reliance on outcrossing reproduction of polyploid variants; and (ii) increased reliance on outcrossing reproduction of diploid variants [27–30].

# 2. Material and methods

## 2.1. Data collection

Data organization and analyses were conducted using the R [31] packages 'raster' [32] and 'tidyverse' [33]. We obtained 49 165 occurrence records of the 72 *Solanum* species sourced from Castañeda-Álvarez *et al*. [1]. These occurrences represent the most threatened and useful wild relatives of *S. tuberosum*, the previously cleaned points were further filtered for those lacking latitudinal and/or

longitudinal information, resulting in a total of 37 032 total occurrence points [1]. The number of occurrences per species are in electronic supplementary material, table S1. Next, the Köppen–Geiger three-tier climate class system was acquired from Rubel & Kottek [34]. The Köppen–Geiger climate class system divides climates into five main groups that are subdivided based on seasonal precipitation and temperature that result in 30 potential discrete classes globally (reviewed in [34]). The Köppen–Geiger is one of the most widely used systems for analysing ecological conditions and identifying primary types of plants of a latitudinal and longitudinal intersection. Descriptions of the climate classes are found in electronic supplementary, table S2, and positions in climate space are found in electronic supplementary material, figure S1. Three-tier climate classes were extracted at each occurrence point. The total number of climate classes per species was counted for each species, and climate classes with three or fewer occurrences were removed in order to avoid 'by-chance' occurrences. Even though climate classes are artificially defined entities without hard boundaries, using discrete climate classes allows for a common measure of both niche diversity and breadth, which is easily operationalized by agricultural professionals. See github repository 'https://github.com/Nfumia/Potato_nichediversity_drivers ' for code and data files.

## 2.2. Ploidy and breeding system data

*Solanum* species ploidy was assessed from Rivero *et al.* [35], where they classified ploidy by curating all the records available from the chromosome count database CCDB [36] and then manually checking for multiplicities of 12 to decide what was a diploid ($2n = 24$ or close to that number in the presence of aneuploidies) or a polyploid ($3x = 36$, $4x = 48$, and so forth). In the original database, populations that have mixed ploidies were recorded (e.g. *Solanum andreanum* is both diploid and polyploid). A recent paper on *Brassica* species by Román-Palacios *et al.* [37] showed that both classifications are mostly indistinguishable in trees with a lot of tips, which is the case of Solanaceae. In our dataset there are higher polyploids ($6x$ and $8x$, e.g. *Solanum demissium* is $6x = 72$). The self-compatibility assessment builds on the data from Goldberg & Igić [38] where new species were added based on the T2-type RNases when data was available after 2012. There are species with both self-incompatible (SI) and self-compatible (SC) populations in the dataset but the level of outcrossing is difficult to tell for all of the species—650 taxa [35].

## 2.3. Linear models for climate classes

A linear model was fitted using R package 'stats' [31] to identify which interaction of biological factors is correlated with niche diversity in *Solanum* section *Petota*. We used the number of discrete climate classes in which each taxon can occur as a proxy for niche breadth, as these niches vary spatially within the five broad descriptors of tropical, dry, temperate, continental and polar, each of which possessing 2–12 subclassifications. For example, the niche of *Solanum stoloniferum* has 15 discrete climate classes, but the niche of *Solanum albornozii* has only one, a temperate oceanic environment. The number of discrete climate classes is the response variable for the model. The predictor variables were combinations of ploidy [1] and breeding system [9,22] for each species, which were coded as dummy variable interaction terms: self-incompatible diploid, self-compatible diploid, self-compatible polyploid and asexually propagating diploid.

## 2.4. Phylogenetic tree and phylogenetic linear models

A Bayesian molecular clock phylogeny with time-calibration of section *Petota* to outgroups of domesticated tomato (*Solanum lycopersicum*) and domesticated eggplant (*Solanum melongena*) was estimated using 32 plastid genomes and compared with the most recent time-calibrated phylogeny of Särkinen *et al.* [39]. Due to a lack of plastid genome availability for some species in *Solanum* section *Petota*, only 27 of the 72 prioritized wild relative species were present in our subsequent analyses. Furthermore, 32 species (27 potato wild relatives, two domesticated potato, one domesticated tomato, one tomato wild relative and one domesticated eggplant) were aligned using the software MAFFT (multiple alignment using fast Fourier transform) via maxiterate version [40]. MrBayes [41] as implemented in the Geneious software package [42] was used to conduct an initial phylogenetic analysis [43,44]. We used a chain length of 10 million generations with 25% (or 2.5 million) burn-in and a subsampling frequency every 1000 generations. The general time reversible (GTR) substitution model was employed for the Bayesian analysis with rate variation of gamma, including four categories.

We used the Bayesian uncorrelated relaxed clock-model dating method as implemented in BEAST2 [45]. The uncorrelated relaxed clock-model allows for rate variation across branches and measures for rate autocorrelation between lineages. Node ages are estimated simultaneously in BEAST2, and, therefore, uncertainty is incorporated into the node-age estimation. Our Bayesian MCMC tree output was used as a starting phylogeny. The Hasegawa–Kishino–Yano (HKY) model for DNA base pair substitution was used to better estimate the substitution rates of transition versus transversion as well as the Felsenstein (F81) proposed four-parameter model. A Kappa of 2.0, as estimated by BEAUti2 [45], was employed. Calibration points for the node-age estimation were sampled from Särkinen *et al.* [39] to create calibration priors: (i) tomato–potato split *ca* 8 Ma (95% HPD 7–10) and (ii) eggplant–tomato/ potato split *ca* 14.3 Ma (95% HPD 13–16). These calibration points reflect a normal distribution with standard deviations of 0.85 and 1.10 Myr, respectively. Yule tree prior with uniform distribution was used given all ingroup and outgroup species in this study that currently persist *ex situ* and/or *in situ*. Priors were manually generated for each monophyletic clade showing greater than 85% posterior probability from the MrBayes MCMC analysis. Default priors were used for all other parameters. A total of 100 million generations, 10 runs with 10 million generations each, were run in BEAST2 [45].

Using the time-calibrated phylogeny (electronic supplementary material, figure S2), we estimated the phylogenetic generalized linear models version of the OLS models proposed in the previous section to account for potential phylogenetic signals in the errors [46,47]. This is an important step, since it is possible that our explanatory variables are not tracking the evolutionary history of the *Petota* section, and can incorrectly conclude strong correlations between the climatic classes and the life-history traits [48].

These phylogenetic linear models were estimated using a maximum-likelihood phylogenetic generalized least-squares (PGLM) with the R package 'phylolm' [49]. For all the PGLMs we assumed a Brownian motion model of evolution [47,50,51]. Outgroup species and cultivated potato were removed at this point due to the inability to differentiate between cultivated and wild occurrence of the given species. This resulted in retention of 27 potato wild relative species, comprising the four major monophyletic clades of section *Petota* [52], for use in the PGLMs analysis.

# 3. Results

## 3.1. Climate regression

The 72 prioritized species in the *Solanum* section *Petota* examined here occurred in 17 distinct climates (where the combination of the discrete climate classes in total make up the climate niche) with individual species distributions ranging from a single climate (e.g. *S. albornozii*, *S. chilliasense*, *S. lesteri*) to 15 distinct climates (e.g. *S. stoloniferum*). The count can, therefore, be a reasonable proxy for niche breadth. Within this range exists a spectrum of breeding system and ploidy combinations between and within these species and their populations, exhibiting different extents of climate niche diversity (figure 1). This analysis showed that distinct breeding system and ploidy combinations existed in a different number of niches ($p = 3.4 \times 10^{-7}$), described as the number of discrete Köppen–Geiger climate classes. Species that possess populations that are self-incompatible diploid and self-compatible polyploid show the greatest mean climate diversity with 11 discrete climate classes (figure 1). Self-incompatible diploid species exhibit a greater average niche diversity when compared with self-compatible diploid species (figure 1). Furthermore, species that contain both diploid and polyploid cytotypes demonstrate greater sustained ecological divergence.

The maximum-likelihood intercept value of ecological niche diversity is $2.81 \pm 1.01$ climate classes. Species existing as self-incompatible diploid or self-compatible polyploid have significantly ($p$-value < 0.01) larger climatic niches by $3.13 \pm 0.73$ and $3.62 \pm 0.79$ discrete climate classes, respectively (table 1). However, other predictor (self-compatible diploid, asexually propagating unknown breeding system diploid) variable slope values are not significantly different from zero, and, therefore, they exert no measurable influence on niche diversity within *Solanum* section *Petota*. Overall, the model explained a moderate amount of variance with an adjusted $R^2$ of 0.39.

## 3.2. Evolutionary climate regression

In the PGLMs fitted using our estimated time-calibrated phylogeny (figure 2), we found an estimated intercept value of $6.43 \pm 1.67$ (table 2). The PGLMs confirmed the correlations of OLS models, with self-incompatible diploid ($3.98 \pm 1.04$) and self-compatible polyploid ($2.57 \pm 0.98$) significantly

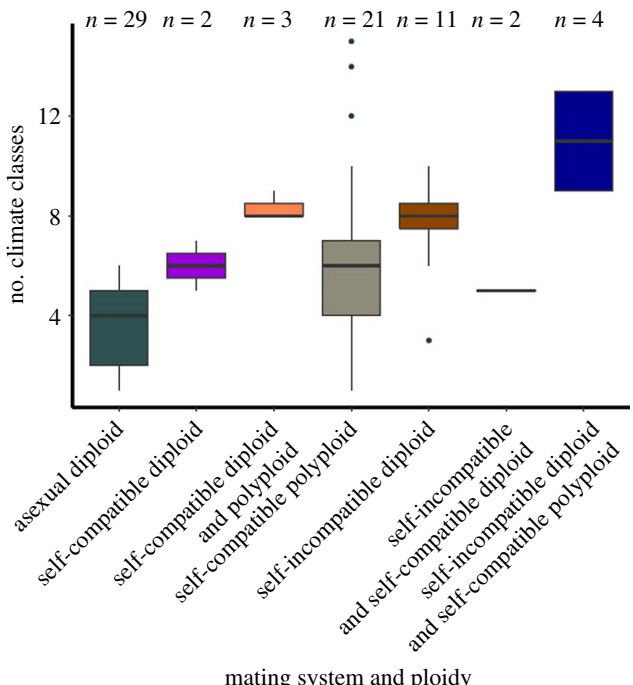

**Figure 1.** Boxplot of niche diversity by breeding system and ploidy interaction in potato wild relative species. Many species exist containing multiple subpopulations with differing biological factors, as seen by combination of such factors on the *x*-axis.

**Table 1.** Results from the linear model for climatic niche diversity following Gaussian distribution. The number of discrete climate classes in which each taxon can occur (i.e. a proxy for niche breadth) is the response variable, climate niche diversity. The predictor variables are combinations of ploidy and breeding system for each species, which were coded as dummy variable interaction terms: self-incompatible diploid, self-compatible diploid, self-compatible asexually propagating polyploid and unknown breeding system asexually propagating diploid. Values reported in column 2 are the maximum-likelihood estimates and standard error of the estimates (surrounded by parentheses).

| | dependent variable: |
|---|---|
| | climatic niche diversity |
| *ordinary least-squares results* | |
| self-incompatible diploid | 3.134*** |
| | (0.734) |
| self-compatible diploid | 0.569 |
| | (1.021) |
| self-compatible polyploid | 3.624*** |
| | (0.789) |
| asexual diploid | 0.883 |
| | (0.992) |
| intercept (MLE) | 2.813*** |
| | (1.006) |
| observations | 72 |
| $R^2$ | 0.426 |
| adjusted $R^2$ | 0.392 |
| residual s.e. | 2.493 (d.f. = 67) |
| *F* statistic | 12.424*** (d.f. = 4; 67) |

***$p < 0.001$.

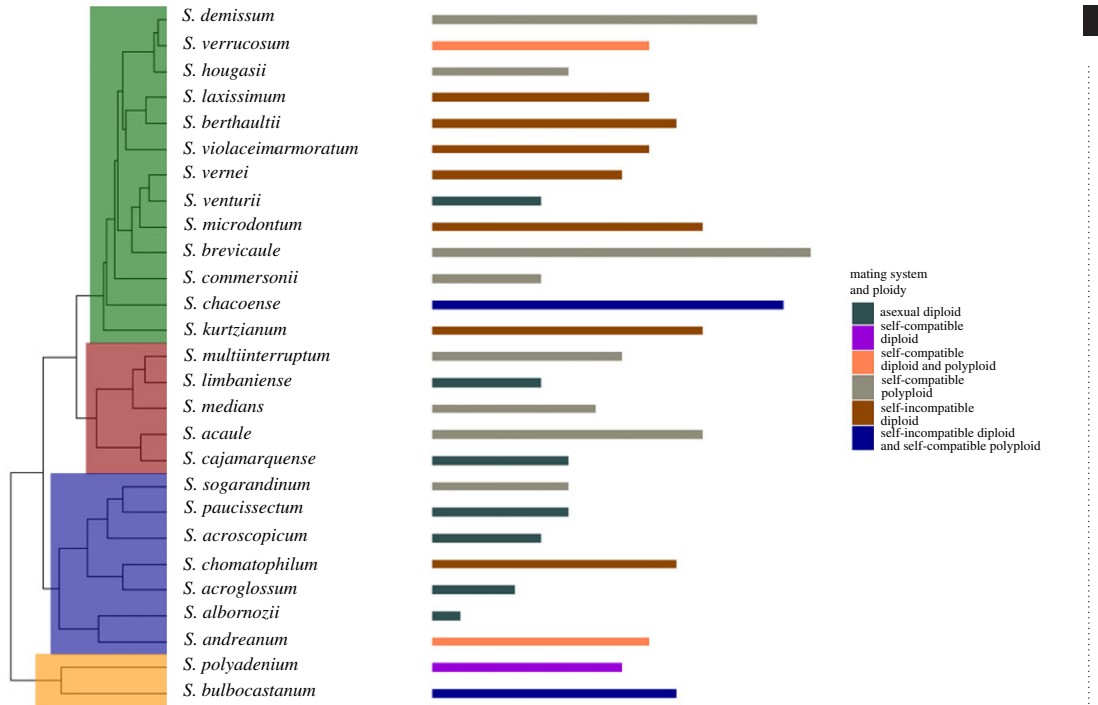

**Figure 2.** Dual figure with time-calibrated molecular clock phylogeny (left) with climatic niche diversity (i.e. number of climate classes occupied) (right). On the left side, the x-axis scale bars represent millions of years and the background coloration of the phylogenetic tree highlights widely accepted clades of *Solanum* section *Petota*. On the right side, the number of climate classes a species occurs in is represented by the size of the horizontal bar and measured with the x-axis scale bar, and the coloration of the horizontal bars represents species biological attributes as breeding system with ploidy.

increasing ecological diversity (table 2). As with OLS, the other predictor variables in PGLMs are not significantly different from zero.

# 4. Discussion

Clarifying the impacts of plant traits on niche divergence is important to understanding the structure of global patterns of biodiversity and evolution in plant lineages [53]. Furthermore, life-history traits can provide clues about the potential resiliency of plants as humans increasingly develop wildlands. However, resilience may be tightly linked with other characteristics. In *Solanum* section *Petota*, the interaction of two specific characters, breeding system and ploidy, explain a large portion of the variation in niche divergence. The models presented here, OLS and PGLMs, explain 39% and 44% ($R^2$), respectively, of the specific measure of climatic variation present within *Solanum* section *Petota* with two alternate ends of the biological spectrum serving as the most significant predictors. On one end, self-incompatible diploid species exhibit the greatest significant correlation to climatic niche diversity within potato wild relatives. Such sustained diversity is probably the result of constant capacity for outcrossing between these species and their subsequent heterogeneous design, fashioning an adaptive and resilient population through long-distance gene flow [54]. Due to the interaction between ploidy and breeding system, self-incompatible diploid species show niche diversity similar to self-compatible polyploid species, confirming the dynamic nature of the Solanaceae system [9,16,22,23]. However, self-fertilizing polyploid species have a short-term advantage as they can colonize new environments with very few individuals.

For all the Solanaceae family, self-incompatible diploidy has been shown to be the ancestral state [22]; they also have faster net diversification compared with all self-compatibles, both diploid and polyploid [55]. The expectation given the success of these lineages in diversification is that self-incompatible diploids should have broader niches; an unexpected result was that self-compatible polyploids diversified in a similar way. Evolutionarily this may be a temporal effect; polyploids are successful in

**Table 2.** Results from phylogenetic linear models for climatic niche diversity following Brownian motion. The number of discrete climate classes in which each taxon can occur (i.e. a proxy for niche breadth) is the response variable, climate niche diversity. The predictor variables are combinations of ploidy and breeding system for each species, which were coded as dummy variable interaction terms: self-incompatible diploid, self-compatible diploid, self-compatible asexually propagating polyploid and unknown breeding system asexually propagating diploid. Values of reported in column 2 are the maximum-likelihood estimates and standard error of the estimates (surrounded by parentheses).

| | *dependent variable:* |
| --- | --- |
| | climatic niche diversity |
| *phylogenetic least-squares results* | |
| self-incompatible diploid | 3.984*** |
| | (1.036) |
| self-compatible diploid | 1.856 |
| | (1.536) |
| self-compatible polyploid | 2.574* |
| | (0.975) |
| asexual diploid | −2.332 |
| | (1.620) |
| intercept (MLE) | 6.426*** |
| | (1.670) |
| sigma$^2$ | $8.088 \times 10^{-9}$ |
| | $(2.967 \times 10^{-9}, 1.067 \times 10^{-8})$ |
| sigma$^2$ error | 4.497 |
| | (1.650, 5.933) |
| observations | 27 |
| $R^2$ | 0.527 |
| adjusted $R^2$ | 0.441 |
| residual s.e. | 4.761 (d.f. = 22) |
| parametric bootstraps | 100 |

$^*p < 0.05$, $^{***}p < 0.001$.

short time scales, and this may explain the success in diversification identified here; however, this study does not disentangle evolutionary timescales. Our results suggest that self-compatible diploids appear evolutionarily transient, and the evolution of self-compatibility appears to occur very rarely without a polyploidy event in Solanaceae.

Self-compatible polyploid species have increased climatic niche diversity, which, given their increased genetic variation and plasticity through additional sets of chromosomes, makes them capable of adaptive and resilient population generation [56]. Polyploidy allows self-fertilizing section *Petota* species to maintain and derive novel diversity typically observed in outcrossing/self-incompatible diploid populations. These differences between breeding system and ploidy with niche diversity provide support for the use of these variable combinations as driving evolutionary forces, with qualitative results (figure 1) being supported by OLS (table 1) and PGLMs (table 2).

Our results suggest the potential to use ecologically plastic species could be useful to enhance the adaptability of cultivated potato lines in the face of climate change. However, the wild species often have limited cross-compatibility with *S. tuberosum*, as evidenced in their endosperm balance numbers, requiring extensive empirical testing. Therefore, time is needed in order to operationalize the use of CWRs in breeding programmes, so that favourable environmental adaptations from a subset of ecologically plastic species, can be introgressed while breaking linkages to agronomically unfavourable traits. Matching individual climate classes with potential places where cropping may shift helps provide a heuristic to match agroclimatic zones with potential donor species, this provides breeders with potentially adapted material for future projected climate shifts.

The impact of breeding system on the evolution of climatic niche breadth among plants is still unclear and the *Solanum* section *Petota* system contributes important evidence for a multilayered role where breeding system and ploidy interact synergistically with one another. In one case, self-incompatible breeding systems play a large role in sustaining niche diversity over time [19] when species are diploid, possessing limited reproductive barriers. By contrast, self-compatible breeding systems comparatively increase niche diversity when species are polyploid, by enhancing their ability to reach, reproduce, establish and adapt [17] with the biological safety net of increased 'buffering capacity' through genetic variation [11]. Further investigations could focus on the decoupling of breeding system and ploidy; however, due to the self-incompatibility conferred by S-RNases found in polyploid populations of Solanaceae this is challenging [9,16,22]. Furthermore, this study was not able to completely decouple ploidy and breeding system interactions due to lack of data on particular species' breeding systems, exemplifying the need for more than DNA collection. Additionally, a limitation of this analysis is the limited number of species available for PGLMs, which was due to a lack of publicly available plastid genome sequence data. There is also a need for future research on exploring the extent of different populations or cytotypes representing the different breeding systems and ploidy combinations, as well as understanding the grey area in mixed compatibility species. This would allow for a complete decoupling of the life-history traits and would also allow for a more nuanced understanding of species. Further, our approach is limited by the current knowledge of taxonomy, if the species within the section *Petota* are revised then the analysis would need to be updated.

Polyploidization has the potential to broaden the climatic niche to a similar extent as would occur with an outcrossing diploid population. The breeding system is the main driver of niche breadth expansion in the self-incompatible diploid populations, while only a secondary contributor in the self-compatible polyploid populations. Despite the biological differences, the resulting niche diversity is not seen in a difference of preferred climate type but rather the extent of climatic diversity (electronic supplementary material, figure S3). Through decoupling geographical range size and niche breadth [26], this study tests classic theory by using a highly diverse, economically important section of plants. Our findings lend credence to the hypothesis that these ecologically plastic responses evolved over millions of years in species with populations of self-incompatible diploids and self-compatible polyploids, and, therefore, these species should be prioritized for conservation and for use to adapt our cultivated varieties to a changing climate.

Data accessibility. Data and relevant code for this research work are available from GitHub: https://github.com/Nfumia/Potato_nichediversity_drivers and the Zenodo repository: https://doi.org/10.5281/zenodo.5773072.

Authors' contributions. N.F.: data curation, formal analysis, investigation, methodology, writing—original draft, writing—review and editing; D.R.: formal analysis, methodology, supervision, writing—review and editing; R.Z.-F.: formal analysis, investigation, methodology, supervision, writing—review and editing; S.P.: data curation, formal analysis, methodology, writing—review and editing; C.K.K.: conceptualization, data curation, methodology, writing—review and editing; M.A.G.: conceptualization, methodology, writing—review and editing; M.B.K.: conceptualization, methodology, project administration, supervision, writing—review and editing.

All authors gave final approval for publication and agreed to be held accountable for the work performed therein.

Competing interests. We declare we have no competing interests.

Funding. We would like to the Hawaii Agriculture Research Center and the University of Hawaii Office of Sustainability for their support of Nathan Fumia through the Sustainable Agriculture Fellowship. Additional funding provided by The College of Tropical Agriculture and Human Resources, University of Hawaii at Manoa; USDA Cooperative State Research, Education and Extension (CSREES), Grant/Award Number: HAW00942-H and HAW08039-H.

Acknowledgements. We would like to thank the Information Technology Systems at the University of Hawai'i at Manoa for computer processing support, and access to data via the Centro Internacional de la Papa. We would like to thank Cornell University for supporting the sabbatical of Dr Michael A. Gore to contribute to this manuscript.

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
