## [Peer Review File · Royal Society Open Science]

Review History

RSOS-211090.R0 (Original submission)

Review form: Reviewer 1

Is the manuscript scientifically sound in its present form?

No

Are the interpretations and conclusions justified by the results?

No

Is the language acceptable?

Yes

Do you have any ethical concerns with this paper?

No

Have you any concerns about statistical analyses in this paper?

No

Recommendation?

Reject

Comments to the Author(s)

The paper by Fumia and co-workers is a good attempt to characterize niche attributes through climate classes and search for signals of phylogenetic interactions with breeding systems and ploidy in species from *Solanum*.

Overall the paper is well-written, the problem, data collection and results are clearly presented and the discussion is nicely addressed (but see minor comments below).

The only major problem I see is in the methods, and the rationale used to estimate or assess niche diversity. The authors use discrete climate classes as a proxy of niche breadth (lines 101-104) and take these as discrete niches (see following example in line 105) in the estimation of niche breadth (lines 168-171) and niche diversity (lines 158-163).

Those are climate subclasses, not discrete niches.

As the authors probably know, the niche of a species represents a collection of biotic and abiotic variables (and the ranges within them) in which the species is found in nature. It can be considered as an ecological space in which the species can survive. Therefore, there is no such as "fifteen discrete niches" (line 105) of a species. One could perhaps subdivide the niche of a species in groups representing traits to extract more information, for example, for alternative cytotypes within a species. But even in such cases the niche of the species is one.

Moreover, climate classes are arbitrary classifications which do not reflect niche differences and cannot be used as a proxy to a species' climatic niche. For example, even within a tropical wet forest there are different niches based on macro-ecology and the species under consideration.

Instead, climate classes are habitat or ecosystem classification of a species.

This implies that, even when conclusions are not wrong, the use of terms related to niche attributes is conceptually wrong or misleading (see e.g., the statement on niche divergence in lines 187-189, or the one about breeding systems and evolution of climatic niche diversity in lines 228-232)

Minor comments

As a minor comment, it is not completely clear why the authors consider the compatibility system when evaluating taxa propagating asexually (lines 107-110). Additionally, since breeding system in asexually propagating diploids is unknown, it makes sense to remove the (self-) compatibility system from the comparison to the asexually propagating polyploids.

I would also be careful about making statements using breeding system (i.e., self-incompatible vs. self-compatible) as discrete classes. The truth is that there are greys in these classes and many species show intermediate levels of pollen-pistil compatibility or variation in the activity of SSI and GSI systems (e.g., unilateral vs. bilateral cross-incompatibility in *Solanum* species).

lines 166-167: a diploid species does not possess polyploids, the species have diploid and polyploid cytotypes, and populations cannot "show polyploidization".

lines 184-187: this sound weird. How the "plants with increased development of wild areas can lead to changes in habitat and climate"?

line 201: "diploidy" or "the self-incompatible diploid state has"

lines 210-211: this is wrong. The breakdown of self-incompatibility is caused by an increase in ploidy, not the other way. At least this is what is known for plants. Many polyploids still conserve self-incompatibility systems.

lines 220-227: this is highly speculative, and I found no need to write such statement in the current study. First, no measure of genetic diversity was estimated nor considered here. Second, *Solanum* species show a complex interaction of genomes during endosperm development (EBN) which render unfruitful many interspecific crosses.

Even in the case the observed variability in species' habitat occurrences presented here is associated to genetic diversity parameters, there is no possible way of concluding this can be transferred to cultivated *Solanum* varieties.

Review form: Reviewer 2

Is the manuscript scientifically sound in its present form?

Yes

Are the interpretations and conclusions justified by the results?

No

Is the language acceptable?

Yes

Do you have any ethical concerns with this paper?

No

Have you any concerns about statistical analyses in this paper?

No

Recommendation?

Major revision is needed (please make suggestions in comments)

Comments to the Author(s)

The study addresses very relevant topic on how polyploidy and breeding system ****interacts**** in order to affect niche breadth of a genus. The question is certainly very interesting, however I am not sure how well-suited the presented model group is given (i) absolute correlation between ploidy and breeding system (polyploids exhibit only one breeding system, self-compatible asexual, which is not present in a diploid state) and lack of own data, specifically sampled to fill in particular gaps in the sampling (e.g. shortage of the plastome data, causing PGLS analysis relying only on 27 out of 72 focal species). A re-analysis using modern statistical methods may certainly also bring novel results, the question is, however, if the dataset is suitable for addressing this question. Please see the following remarks.

1) Unfortunately, it is totally unclear how the data were acquired, especially how the breeding system information and ploidy has been assigned. Do the authors simply fully rely on the previously published works cited in Methods? Even if so it would be worth explaining how the data have been acquired in the original works and potentially critically review their approach. Or did the authors curated the data anyhow, re-assigning the group or updating with potential novel data?. Specifically:

- How the ploidy was assigned to a each species? What is the basic chromosome number for *Solanum* and which $2n=$ value is thus assigned to be a diploid and tetraploid? In how many cases more chromosome counts have been recorded per species and in what proportion of such cases intraspecific ploidy variation has been observed (e.g. diploid and tetraploid chromosome counts in one species)? How ploidy-variable species were then handled (excluded / assigned to both classes)? Are there higher polyploids than tetraploids in the analysis?

- How was self-compatibility assessed and does this trait also vary within some species (it can do, e.g. in *Arabidopsis lyrata*). If so, how such variable species have been handled? Is there anything

known about the outcrossing levels of the SC species (i.e. real manifestation of the breeding system shift in natural populations)?

2) The absolute correlation of ploidy and breeding system (polyploids are always self-compatible asexuals, and this combination of breeding systems is not present in a diploid, l. 108-110) makes impossible to decouple the effect of ploidy and breeding system. This is the nature and it is worth studying, yet under the current natural setup it is impossible to distill the role of ploidy in niche evolution (contrary e.g. to the statements on l. 196). Re-focusing the main aim towards breeding system (taking ploidy into account as a potential side-effect), or the study and/or focusing on different model system may help to address this issue. In the former case, however, I am not sure about the novelty of the study as relatively a lot of work has been done on the relationships between breeding systems and distributions.

Minors:

What is "unknown breeding system asexually propagating diploid"? Is it asexual or unknown?

Even putting polyploidy apart, it is also unclear to which extent the expansion of the "self compatible asexual polyploid" category is due to selfing and asexuality. This shall be also taken into account in the Discussion (e.g. 208-219)

Decision letter (RSOS-211090.R0)

Dear Dr Kantar

The Editors assigned to your paper RSOS-211090 "Interactions between breeding system and ploidy affect niche breadth in *Solanum*" have made a decision based on their reading of the paper and any comments received from reviewers.

Regrettably, in view of the reports received, the manuscript has been rejected in its current form. However, a new manuscript may be submitted which takes into consideration these comments.

We invite you to respond to the comments supplied below and prepare a resubmission of your manuscript. Below the referees' and Editors' comments (where applicable) we provide additional requirements. We provide guidance below to help you prepare your revision.

Please note that resubmitting your manuscript does not guarantee eventual acceptance, and we do not generally allow multiple rounds of revision and resubmission, so we urge you to make every effort to fully address all of the comments at this stage. If deemed necessary by the Editors, your manuscript will be sent back to one or more of the original reviewers for assessment. If the original reviewers are not available, we may invite new reviewers.

Please resubmit your revised manuscript and required files (see below) no later than 21-Apr-2022. Note: the ScholarOne system will 'lock' if resubmission is attempted on or after this deadline. If you do not think you will be able to meet this deadline, please contact the editorial office immediately.

Please note article processing charges apply to papers accepted for publication in Royal Society Open Science (<https://royalsocietypublishing.org/rsos/charges>). Charges will also apply to papers transferred to the journal from other Royal Society Publishing journals, as well as papers submitted as part of our collaboration with the Royal Society of Chemistry (<https://royalsocietypublishing.org/rsos/chemistry>). Fee waivers are available but must be requested when you submit your manuscript (<https://royalsocietypublishing.org/rsos/waivers>).

Thank you for submitting your manuscript to Royal Society Open Science and we look forward to receiving your resubmission. If you have any questions at all, please do not hesitate to get in touch.

on behalf of Dr Joachim Mergeay (Associate Editor) and Pete Smith (Subject Editor)
openscience@royalsociety.org

Associate Editor Comments to Author (Dr Joachim Mergeay):

Comments to the Author:

We've now had two reviews on your manuscript. Both reviewers raise very different but pertinent issues: one disputes that a) the definition of a climatic niche is adequate and b) that climate zones are not categorical entities, and therefore the manuscript does not adequately represent an analysis of niche breadth. This seems more than a semantic discussion on the term 'niche'.

The second reviewer highlights that ploidy and mating system co-vary to a large extent, as a result of which a decoupling between ploidy and mating system is not possible (line 109: all polyploids are self-compatible asexuals).

As for myself, I had a hard time understanding some concepts. How is a an asexual species self-compatible? If it's asexual, self-incompatibility is irrelevant, no? If on the other hand it depends on pollination, it's not asexually propagating. This conundrum shows that much clearer definitions of what is understood by the different mating systems are required in the text, as we are not speaking the same language here.

Next, you talk about polyploid versus diploid species, yet you also consider "species" having both asexual polyploids, and sexual diploids. To what extent is this sensitive to "advances in taxonomic insight?" When asexuality and polyploidy are involved, the boundaries of species are often arbitrary and a matter of convention, rather than representing clear biological or evolutionary units. Hence, are all 'species' equivalent? What may be considered a single species with a large variation in ploidy and mating systems, may already have been split into various distinct species in another taxon. To what extent is this analysis sensitive to such taxonomic idiosyncracies?

One may even argue to what extent it is useful to consider evolutionary distinct entities together (polyploid vs diploid, sexual vs asexual), even though they are considered as part of the same taxonomic man-made box. Do the polyploids in a 'species' containing both diploid and polyploid lineages have a larger niche breadth than the polyploids in a 'species' containing only polyploids? And is this niche diversity dependent on the sample size (was a same number of lineages considered for each group?). Each asexual lineage could have its own narrow niche (and the combination of all lineages may show a broad niche). Alternatively, asexual polyploid lineages

may indeed have broad niches. This is a classical question in the evolutionary biology and ecology of asexuality.

Overall, there are a few fundamental issues that need to be resolved and clarified before this work can be considered any further.

I am sorry to disappoint you with this decision, but hope it will help you improve a revised manuscript.

Sincerely,
Joachim Mergeay

Reviewer comments to Author:

Reviewer: 1

Comments to the Author(s)

The paper by Fumia and co-workers is a good attempt to characterize niche attributes through climate classes and search for signals of phylogenetic interactions with breeding systems and ploidy in species from *Solanum*.

Overall the paper is well-written, the problem, data collection and results are clearly presented and the discussion is nicely addressed (but see minor comments below).

The only major problem I see is in the methods, and the rationale used to estimate or assess niche diversity. The authors use discrete climate classes as a proxy of niche breadth (lines 101-104) and take these as discrete niches (see following example in line 105) in the estimation of niche breadth (lines 168-171) and niche diversity (lines 158-163).

Those are climate subclasses, not discrete niches.

As the authors probably know, the niche of a species represents a collection of biotic and abiotic variables (and the ranges within them) in which the species is found in nature. It can be considered as an ecological space in which the species can survive. Therefore, there is no such as "fifteen discrete niches" (line 105) of a species. One could perhaps subdivide the niche of a species in groups representing traits to extract more information, for example, for alternative cytotypes within a species. But even in such cases the niche of the species is one.

Moreover, climate classes are arbitrary classifications which do not reflect niche differences and cannot be used as a proxy to a species' climatic niche. For example, even within a tropical wet forest there are different niches based on macro-ecology and the species under consideration. Instead, climate classes are habitat or ecosystem classification of a species.

This implies that, even when conclusions are not wrong, the use of terms related to niche attributes is conceptually wrong or misleading (see e.g., the statement on niche divergence in lines 187-189, or the one about breeding systems and evolution of climatic niche diversity in lines 228-232)

Minor comments

As a minor comment, it is not completely clear why the authors consider the compatibility system when evaluating taxa propagating asexually (lines 107-110). Additionally, since breeding system in asexually propagating diploids is unknown, it makes sense to remove the (self-) compatibility system from the comparison to the asexually propagating polyploids.

I would also be careful about making statements using breeding system (i.e., self-incompatible vs. self-compatible) as discrete classes. The truth is that there are greys in these classes and many species show intermediate levels of pollen-pistil compatibility or variation in the activity of SSI and GSI systems (e.g., unilateral vs. bilateral cross-incompatibility in *Solanum* species).

lines 166-167: a diploid species does not possess polyploids, the species have diploid and polyploid cytotypes, and populations cannot "show polyploidization".

lines 184-187: this sound weird. How the "plants with increased development of wild areas can lead to changes in habitat and climate"?

line 201: "diploidy" or "the self-incompatible diploid state has"

lines 210-211: this is wrong. The breakdown of self-incompatibility is caused by an increase in ploidy, not the other way. At least this is what is known for plants. Many polyploids still conserve self-incompatibility systems.

lines 220-227: this is highly speculative, and I found no need to write such statement in the current study. First, no measure of genetic diversity was estimated nor considered here. Second, *Solanum* species show a complex interaction of genomes during endosperm development (EBN) which render unfruitful many interspecific crosses.

Even in the case the observed variability in species' habitat occurrences presented here is associated to genetic diversity parameters, there is no possible way of concluding this can be transferred to cultivated *Solanum* varieties.

Reviewer: 2

Comments to the Author(s)

The study addresses very relevant topic on how polyploidy and breeding system ****interacts**** in order to affect niche breadth of a genus. The question is certainly very interesting, however I am not sure how well-suited the presented model group is given (i) absolute correlation between ploidy and breeding system (polyploids exhibit only one breeding system, self-compatible asexual, which is not present in a diploid state) and lack of own data, specifically sampled to fill in particular gaps in the sampling (e.g. shortage of the plastome data, causing PGLS analysis relying only on 27 out of 72 focal species). A re-analysis using modern statistical methods may certainly also bring novel results, the question is, however, if the dataset is suitable for addressing this question. Please see the following remarks.

1) Unfortunately, it is totally unclear how the data were acquired, especially how the breeding system information and ploidy has been assigned. Do the authors simply fully rely on the previously published works cited in Methods? Even if so it would be worth explaining how the data have been acquired in the original works and potentially critically review their approach. Or did the authors curated the data anyhow, re-assigning the group or updating with potential novel data?. Specifically:

- How the ploidy was assigned to a each species? What is the basic chromosome number for *Solanum* and which $2n=$ value is thus assigned to be a diploid and tetraploid? In how many cases more chromosome counts have been recorded per species and in what proportion of such cases intraspecific ploidy variation has been observed (e.g. diploid and tetraploid chromosome counts in one species)? How ploidy-variable species were then handled (excluded / assigned to both classes?)? Are there higher polyploids than tetraploids in the analysis?

- How was self-compatibility assessed and does this trait also vary within some species (it can do, e.g. in *Arabidopsis lyrata*). If so, how such variable species have been handled? Is there anything known about the outcrossing levels of the SC species (i.e. real manifestation of the breeding system shift in natural populations)?

2) The absolute correlation of ploidy and breeding system (polyploids are always self-compatible asexuals, and this combination of breeding systems is not present in a diploid, l. 108-110) makes impossible to decouple the effect of ploidy and breeding system. This is the nature and it is worth studying, yet under the current natural setup it is impossible to distill the role of ploidy in niche evolution (contrary e.g. to the statements on l. 196). Re-focusing the main aim towards breeding system (taking ploidy into account as a potential side-effect), or the study and/or focusing on different model system may help to address this issue. In the former case, however, I am not sure about the novelty of the study as relatively a lot of work has been done on the relationships between breeding systems and distributions.

Minors:

What is "unknown breeding system asexually propagating diploid"? Is it asexual or unknown?

Even putting polyploidy apart, it is also unclear to which extent the expansion of the "self compatible asexual polyploid" category is due to selfing and asexuality. This shall be also taken into account in the Discussion (e.g 208-219)

===PREPARING YOUR MANUSCRIPT===

===PREPARING YOUR REVISION IN SCHOLARONE===

Please ensure that you include a summary of your paper at Step 2 'Type, Title, & Abstract'. This should be no more than 100 words to explain to a non-scientific audience the key findings of your

research. This will be included in a weekly highlights email circulated by the Royal Society press office to national UK, international, and scientific news outlets to promote your work.

Author's Response to Decision Letter for (RSOS-211090.R0)

See Appendix A.

Decision letter (RSOS-211862.R0)

Dear Dr Kantar

On behalf of the Editors, we are pleased to inform you that your Manuscript RSOS-211862 "Interactions between breeding system and ploidy affect niche breadth in *Solanum*" has been accepted for publication in Royal Society Open Science subject to minor revision in accordance with the referees' reports. Please find the referees' comments along with any feedback from the Editors below my signature.

Please submit your revised manuscript and required files (see below) no later than 7 days from today's (ie 03-Dec-2021) date. Note: the ScholarOne system will 'lock' if submission of the revision is attempted 7 or more days after the deadline. If you do not think you will be able to meet this deadline please contact the editorial office immediately.

on behalf of Dr Joachim Mergeay (Associate Editor) and Pete Smith (Subject Editor)
openscience@royalsociety.org

Associate Editor Comments to Author (Dr Joachim Mergeay):

Dear authors,

I went through the revised manuscript, and added a few extra notes as comments in the pdf. Overall, I understand that you categorize climate into distinct categories (classes) for analytical reasons, but these are still no real physical entities with clear boundaries. This still needs to be highlighted more clearly at the onset of the methods (§ data collection).

Ensure that figures are intelligible (also supplementary). This typically means increasing font sizes, increasing line widths, and adapt color schemes in R to something a bit more color-blind

friendly, or at least with more contrast between adjacent colors. There were so many different hues of green, blue, reddish, pink in some supplemental figures that they became hard to understand (suppl figs 1 and 3 in particular). Also, please explain abbreviations in supplemental figures too.

Sincerely,
Joachim Mergeay
Associate editor

===PREPARING YOUR MANUSCRIPT===

one version should clearly identify all the changes that have been made (for instance, in coloured highlight, in bold text, or tracked changes);

===PREPARING YOUR REVISION IN SCHOLARONE===

-- If you are requesting an article processing charge waiver, you must select the relevant waiver option (if requesting a discretionary waiver, the form should have been uploaded, see 'File upload' above).

-- If you have uploaded any electronic supplementary (ESM) files, please ensure you follow the guidance at <https://royalsociety.org/journals/authors/author-guidelines/#supplementary-material> to include a suitable title and informative caption. An example of appropriate titling and captioning may be found at https://figshare.com/articles/Table_S2_from_Is_there_a_trade-off_between_peak_performance_and_performance_breadth_across_temperatures_for_aerobic_scope_in_teleost_fishes_/3843624.

Author's Response to Decision Letter for (RSOS-211862.R0)

See Appendix B.

Decision letter (RSOS-211862.R1)

Dear Dr Kantar,

I am pleased to inform you that your manuscript entitled "Interactions between breeding system and ploidy affect niche breadth in *Solanum*" is now accepted for publication in Royal Society Open Science.

The proof of your paper will be available for review using the Royal Society online proofing system and you will receive details of how to access this in the near future from our production office (opencscience_proofs@royalsociety.org). We aim to maintain rapid times to publication after acceptance of your manuscript and we would ask you to please contact both the production office and editorial office if you are likely to be away from e-mail contact to minimise delays to publication. If you are going to be away, please nominate a co-author (if available) to manage the proofing process, and ensure they are copied into your email to the journal.

Kind regards,
Royal Society Open Science Editorial Office
Royal Society Open Science
opencscience@royalsociety.org

on behalf of Dr Joachim Mergeay (Associate Editor) and Pete Smith (Subject Editor)
opencscience@royalsociety.org

Appendix A

Response to reviewer comments

Editor

Comment: The definition of a climatic niche is not adequate and b) that climate zones are not categorical entities, and therefore the manuscript does not adequately represent an analysis of niche breadth. This seems more than a semantic discussion on the term 'niche'.

Response: We have reframed the discussion of climate niche into a more applied agricultural focus, here we use the discrete climate classes to understand the way plants have moved between human defined systems and how this may provide increased use in plant breeding and the implications for applied conservation.

Comment: ploidy and mating system co-vary to a large extent, as a result of which a decoupling between ploidy and mating system is not possible (line 109: all polyploids are self-compatible asexuals).

Response: While there is a large amount of co-variation, there is also signal of main effects of the ploidy and mating system. So, we do see some decoupling due to these main effects and this has great utility both for explaining niche expansion and potential use in breeding.

Comment: How is an asexual species self-compatible? If it's asexual, self-incompatibility is irrelevant, no? If on the other hand it depends on pollination, it's not asexually propagating. This conundrum shows that much clearer definitions of what is understood by the different mating systems are required in the text, as we are not speaking the same language here.

Response: Reproduction can be self-compatible and/or asexual. They are reinforcing mechanisms. In the case of the polyploid species explored here they completely covary- we gain some intuition through the model but there is more work to be done. We have added additional caveats and tried to explain why the intuition we gain in this instance can be helpful for understanding ecological theory and plant breeding.

Comment: you talk about polyploid versus diploid species, yet you also consider "species" having both asexual polyploids, and sexual diploids. To what extent is this sensitive to "advances in taxonomic insight?"

Response: We are using the most current taxonomy available for the species, some species have been shown to have a ploidy series. However, this is a limitation and we have added it as a caveat (line 261-267). While there may be a few taxonomic revisions to come this is a multi-species analysis and most of the taxa are relatively well-resolved, permitting broader conclusions. Also, there is not much advance to taxonomic insight as PGLS is pretty robust under simple changes of the phylogeny. Therefore, significant changes to the tree would be necessary to change the result.

Comment: When asexuality and polyploidy are involved, the boundaries of species are often arbitrary and a matter of convention, rather than representing clear biological or evolutionary units. Hence, are all 'species' equivalent?

Response:

This is a concern, but without redefining the taxonomy we cannot weight different species. We have added this as a caveat. The species also have fairly similar growth habit, this implies that there is fair comparison between species in different environments. Further, the question of “species” is a far-ranging and intractable one contingent on varying definitions of ‘species’, which would be true for any study using species level taxa, certainly not limited to the one we present here. By following conventions and being clear about the provenance and status of each species we use, the results we have generated will apply regardless of future nomenclatural changes. Furthermore, we are using the phylogenetic species (genealogical) as described in de Queiroz (2007). Under this definition, all alleles of a given gene have descended from a common ancestral allele and are not shared with those of other species. It definitely applies to polyploids, because the duplicate alleles descend from the same gene that is ancestor to the diploid population. We are not using the biological concept of species here because that is problematic. We also have assume autopolyploidy, and not allopolyploidy which would require phylogenetic networks not trees.

De Queiroz, K. (2007). Species concepts and species delimitation. *Systematic biology*, 56(6), 879-886.

Comment: What may be considered a single species with a large variation in ploidy and mating systems, may already have been split into various distinct species in another taxon.

Response: In this case we used the current taxonomic classification, we cannot analyze future potential cases, but we can explore current hypothesis with the current taxonomy. This is also the case for any group in which species level research is being conducted. And it’s possible that our results might stimulate finer taxonomic studies of species which exhibit broad ranges. But we are using the most up to date species identifications for this research. It would be a very high standard if every species-level study had to first examine and revise the taxonomy of the group in question. Again, by providing data on the provenance of each OTU we use, the reproducibility of our results by future researchers is facilitated. Additionally, the SOL source spent significant amount of time delimiting species, we are using their taxonomical description, these are well known. <https://solanaceaesource.myspecies.info/>

Comment: To what extent is this analysis sensitive to such taxonomic idiosyncracies?

One may even argue to what extent it is useful to consider evolutionary distinct entities together (polyploid vs diploid, sexual vs asexual), even though they are considered as part of the same taxonomic man-made box. Do the polyploids in a 'species' containing both diploid and polyploid lineages have a larger niche breadth than the polyploids in a 'species' containing only polyploids? And is this niche diversity dependent on the sample size (was a same number of lineages considered for each group?). Each asexual lineage could have its own narrow niche (and the combination of all lineages may show a broad niche). Alternatively, asexual polyploid lineages may indeed have broad niches. This is a classical question in the evolutionary biology and

ecology of asexuality.

Response: It is well established that potato species contain multiple cytotypes, therefore multiple ploids exist in a single species, often with increased originating multiple times (Hijmans et al., 2007; Spooner et al., 2010).

Hijmans, R. J., Gavrilenko, T., Stephenson, S., Bamberg, J., Salas, A., & Spooner, D. M. (2007). Geographical and environmental range expansion through polyploidy in wild potatoes (*Solanum* section *Petota*). *Global ecology and biogeography*, 16(4), 485-495.

Spooner, D. M., Gavrilenko, T., Jansky, S. H., Ovchinnikova, A., Krylova, E., Knapp, S., & Simon, R. (2010). Ecogeography of ploidy variation in cultivated potato (*Solanum* sect. *Petota*). *American journal of botany*, 97(12), 2049-2060.

Reviewer 1

Comments: The paper by Fumia and co-workers is a good attempt to characterize niche attributes through climate classes and search for signals of phylogenetic interactions with breeding systems and ploidy in species from *Solanum*.

Overall the paper is well-written, the problem, data collection and results are clearly presented and the discussion is nicely addressed (but see minor comments below).

Response: Thank you.

Comment: The only major problem I see is in the methods, and the rationale used to estimate or assess niche diversity. The authors use discrete climate classes as a proxy of niche breadth (lines 101-104) and take these as discrete niches (see following example in line 105) in the estimation of niche breadth (lines 168-171) and niche diversity (lines 158-163). Those are climate subclasses, not discrete niches. As the authors probably know, the niche of a species represents a collection of biotic and abiotic variables (and the ranges within them) in which the species is found in nature. It can be considered as an ecological space in which the species can survive. Therefore, there is no such as "fifteen discrete niches" (line 105) of a species. One could perhaps subdivide the niche of a species in groups representing traits to extract more information, for example, for alternative cytotypes within a species. But even in such cases the niche of the species is one.

Response: We have reframed this discussion around operationalizing the use of species in broad vs. narrow niches. Further, we have also reframed the discussion of the climate classes being a measure of the breadth as opposed to a discrete niche. While other factors besides climate will dictate niche, it is perhaps one of the most important factors in niche determination.

Comment: Moreover, climate classes are arbitrary classifications which do not reflect niche differences and cannot be used as a proxy to a species' climatic niche. For example, even within a tropical wet forest there are different niches based on macro-ecology and the species under consideration. Instead, climate classes are habitat or ecosystem classification of a species. This implies that, even when conclusions are not wrong, the use of terms related to niche attributes is

conceptually wrong or misleading (see e.g., the statement on niche divergence in lines 187-189, or the one about breeding systems and evolution of climatic niche diversity in lines 228-232)

Response: We have clarified that climate classes are a measure of the total niche that allows for more specific decisions regarding sourcing populations for breeding and conservation. This clarification has to do with the terminology using the discrete classes to discuss niche breadth rather than total niche diversity.

Comment: it is not completely clear why the authors consider the compatibility system when evaluating taxa propagating asexually (lines 107-110). Additionally, since breeding system in asexually propagating diploids is unknown, it makes sense to remove the (self-) compatibility system from the comparison to the asexually propagating polyploids.

Response: We have clarified the way species were classified (see new section in methods) and how to interpret these classifications.

Comment: I would also be careful about making statements using breeding system (i.e., self-incompatible vs. self-compatible) as discrete classes. The truth is that there are greys in these classes and many species show intermediate levels of pollen-pistil compatibility or variation in the activity of SSI and GSI systems (e.g., unilateral vs. bilateral cross-incompatibility in *Solanum* species).

Response: We agree that this is a very true biological fact and have added it as a caveat.

Comment: lines 166-167: a diploid species does not possess polyploids, the species have diploid and polyploid cytotypes, and populations cannot “show polyploidization”.

Response: We have changed the sentence to “Furthermore, species that contain both diploid and polyploid cytotypes demonstrate sustained ecological divergence.”

Comment: lines 184-187: this sound weird. How the "plants with increased development of wild areas can lead to changes in habitat and climate"?

Response: We have changed the sentence to read “Furthermore, life history traits can provide clues about the potential resiliency of plants as humans increasingly develop wildlands.”

Comment: line 201: “diploidy” or “the self-incompatible diploid state has”

Response: This was changed to “diploidy”

Comment: lines 210-211: this is wrong. The breakdown of self-incompatibility is caused by an increase in ploidy, not the other way. At least this is what is known for plants. Many polyploids still conserve self-incompatibility systems.

Response: We deleted the sentence “This suggests that polyploidy is just an evolutionary byproduct of trying to become self-compatible, allowing for rapid establishment in many new environments.”

Comment: lines 220-227: this is highly speculative, and I found no need to write such statement in the current study. First, no measure of genetic diversity was estimated nor considered here. Second, *Solanum* species show a complex interaction of genomes during endosperm development (EBN) which render unfruitful many interspecific crosses.

Response: We agree. We have clarified that promising wild relatives require empirical testing.

Comment: Even in the case the observed variability in species' habitat occurrences presented here is associated to genetic diversity parameters, there is no possible way of concluding this can be transferred to cultivated *Solanum* varieties.

Response: While breeding populations are not investigated here, there is a long history in potato breeding and plant breeding generally of using exotic breeding material of different ploidy and species (Jansky et al., 2006; Dempwold et al., 2017), here we attempt to present a way of choosing sources of such material in a way that narrows the number of populations and species that a breeder would have to focus on. We agree that the next step is creating such populations to validate their utility.

Jansky, S. H., Simon, R., & Spooner, D. M. (2006). A test of taxonomic predictivity: Resistance to white mold in wild relatives of cultivated potato. *Crop Science*, 46(6), 2561-2570.

Dempewolf, H., Baute, G., Anderson, J., Kilian, B., Smith, C., & Guarino, L. (2017). Past and future use of wild relatives in crop breeding. *Crop Science*, 57(3), 1070-1082.

Reviewer 2:

Comment: The study addresses very relevant topic on how polyploidy and breeding system ****interacts**** in order to affect niche breadth of a genus. The question is certainly very interesting, however I am not sure how well-suited the presented model group is given (i) absolute correlation between ploidy and breeding system (polyploids exhibit only one breeding system, self-compatible asexual, which is not present in a diploid state) and lack of own data, specifically sampled to fill in particular gaps in the sampling (e.g. shortage of the plastome data, causing PGLS analysis relying only on 27 out of 72 focal species). A re-analysis using modern statistical methods may certainly also bring novel results, the question is, however, if the dataset is suitable for addressing this question. Unfortunately, it is totally unclear how the data were acquired, especially how the breeding system information and ploidy has been assigned. Do the authors simply fully rely on the previously published works cited in Methods?

Response: We have added more information on the data set in the new methods section (line 103-115).

Comment: How the ploidy was assigned to a each species? What is the basic chromosome number for *Solanum* and which $2n=$ value is thus assigned to be a diploid and tetraploid? In how many cases more chromosome counts have been recorded per species and in what proportion of such cases intraspecific ploidy variation has been observed (e.g. diploid and tetraploid chromosome counts in one species)? How ploidy-variable species were then handled (excluded /

assigned to both classes?)? Are there higher polyploids than tetraploids in the analysis?

Response: R-Solanum has a $2n=24$ (haploid $n=12$) classification was done by curating all the records available from the chromosome count database CCDB (citation from Mayrose and others in his lab) using partially software to curate the chromosome records (CCDB curator- Rivero, Sessa, and Zenil-Ferguson) and then manually checking for multiplicities or approximate multiplicities of 12 to decide what was a diploid ($2n=24$ or close to that number in the presence of aneuploidies) or a polyploid ($3x=36$, $4x=48$, and so forth). In the original database populations that have mixed ploidies were recorded (e.g. *Solanum andreanum* is both diploid and polyploid). This is the standard way of classifying although there are other approaches like following the evolutionary history of chromosome numbers like (chromevol-Mayrose) and deciding ploidies based on potential chromosome doublings in ancestral reconstruction. A recent paper in Brassicaceae by Roman-Palacios et al (2020) showed that both classifications are mostly indistinguishable in trees with a lot of tips which is the case of Solanaceae. These ploidy classifications and curation procedure was published by Zenil-Ferguson et al. 2019, and are publicly available. In our dataset there are higher polyploids ($6x$ and $8x$) \ (e.g. *Solanum demissum* is a $6x=72$ and that is classified as a polyploid). Although most polyploids in the Solanum clade are polyploid would be $4x$.

Glick, L., & Mayrose, I. (2014). ChromEvol: assessing the pattern of chromosome number evolution and the inference of polyploidy along a phylogeny. *Molecular biology and evolution*, 31(7), 1914-1922.

Rice, A., Glick, L., Abadi, S., Einhorn, M., Kopelman, N. M., Salman- Minkov, A., et al. (2015). The Chromosome Counts Database (CCDB)—a community resource of plant chromosome numbers. *New Phytologist*, 206(1), 19-26.

Rivero, R., Sessa, E. B., & Zenil- Ferguson, R. (2019). EyeChrom and CCDB curator: Visualizing chromosome count data from plants. *Applications in plant sciences*, 7(1), e01207.

Román-Palacios, C., Molina-Henao, Y. F., & Barker, M. S. (2020). Polyploids increase overall diversity despite higher turnover than diploids in the Brassicaceae. *Proceedings of the Royal Society B*, 287(1934), 20200962.

Zenil- Ferguson, R., Burleigh, J.G., Freyman, W.A., Igić, B., Mayrose, I. and Goldberg, E.E., 2019. Interaction among ploidy, breeding system and lineage diversification. *New Phytologist*, 224(3), pp.1252-1265

Zenil-Ferguson, Rosana et al. (2019), Data from: Interaction among ploidy, breeding system, and lineage diversification, Dryad, Dataset, <https://doi.org/10.5061/dryad.tb7055f>

Comment: How was self-compatibility assessed and does this trait also vary within some species (it can do, e.g. in *Arabidopsis lyrata*). If so, how such variable species have been handled? Is there anything known about the outcrossing levels of the SC species (i.e. real manifestation of the breeding system shift in natural populations)?

Response: SC assessment: In order to binarily classify breeding system for use in macroevolutionary models, each species must either be self-incompatible (SI) or self-compatible (SC). With the commonality of breakdown of SI, there is sometimes the presence of SC mutants, SC populations, or even sister species and therefore some species were encoded as SI where the species had not entirely transitioned to SC (Goldberg et al 2010). But, some were encoded as SC because the expectation of transfer from SI to SC typically precedes gender dimorphism (Miller

and Venable 2000). Where data was originally sourced from Goldberg et al 2010, the final count of SI and SC did not significantly differ from the total available state data.

Miller, J. S., & Venable, D. L. (2000). Polyploidy and the evolution of gender dimorphism in plants. *Science*, 289(5488), 2335-2338.

Goldberg, E. E., Kohn, J. R., Lande, R., Robertson, K. A., Smith, S. A., & Igić, B. (2010). Species selection maintains self-incompatibility. *Science*, 330(6003), 493-495.

Goldberg, E. E., & Igić, B. (2012). Tempo and mode in plant breeding system evolution. *Evolution: International Journal of Organic Evolution*, 66(12), 3701-3709.

To the Goldberg and Igić dataset we added species from recently published articles in the Solanaceae source (after 2012)

<https://solanaceaesource.myspecies.info/taxonomy/term/99332/literature>. In Solanaceae, the standard way to figure out incompatibility is by looking into the presence of T2-type RNases for each species. Boris Igić has done this in his lab very successfully (and other places like USDA), those data were added by Boris himself. The convergent evolution of incompatibility via T2-Type RNases was thoroughly shown by Igić and Kohn (2001

<https://www.pnas.org/content/98/23/13167>) and RNases are the standard to understand incompatibility. A whole review on the topic of RNases as the model system for investigating SI is in McClure, B., 2009. (Darwin's foundation for investigating self-incompatibility and the progress toward a physiological model for S-RNase-based SI. *Journal of experimental botany*, 60(4), pp.1069-1081). There are species with both SI and SC populations in the dataset but the level of outcrossing is difficult to tell for the huge number of species (650 taxa)

Igić, B., & Kohn, J. R. (2001). Evolutionary relationships among self-incompatibility RNases. *Proceedings of the National Academy of Sciences*, 98(23), 13167-13171.

McClure, B. (2009). Darwin's foundation for investigating self-incompatibility and the progress toward a physiological model for S-RNase-based SI. *Journal of experimental botany*, 60(4), 1069-1081

Comment: The absolute correlation of ploidy and breeding system (polyploids are always self-compatible asexuals, and this combination of breeding systems is not present in a diploid, l. 108-110) makes it impossible to decouple the effect of ploidy and breeding system. This is the nature and it is worth studying, yet under the current natural setup it is impossible to distill the role of ploidy in niche evolution (contrary e.g. to the statements on l. 196). Re-focusing the main aim towards breeding system (taking ploidy into account as a potential side-effect), or the study and/or focusing on different model system may help to address this issue. In the former case, however, I am not sure about the novelty of the study as relatively a lot of work has been done on the relationships between breeding systems and distributions.

Response: Polyploids will always be SC but there are SC species that are diploids as well. SI will always be diploids, but diploids can be SI or SC, so there is a partial decoupling thanks to the presence of compatible diploids. This tells us that polyploidy is only one way that incompatibility could be broken but there are others. One way to obtain breeding system shifts is by polyploidy but not the only way. So, the distributions themselves of breeding systems and their difference might be also due to ploidy differences.

Comment: What is "unknown breeding system asexually propagating diploid"? Is it asexual or unknown?

Response: It is both, all of the solanum section petoa have underground storage and are therefore capable of asexually propagating, but some are under surveyed, so the sexual reproduction system is unknown. We clarified this in the text in the new methods section (line 103-115).

Comment: Even putting polyploidy apart, it is also unclear to which extent the expansion of the "self compatible asexual polyploid" category is due o selfing and asexuality. This shall be also taken into account in the Discussion (e.g 208-219)

Response: We agree and have added this as a limitation (line 261-267).

Appendix B

Response to reviewer comments

Editor

Thank you for the helpful comments and suggestions. We are excited to send this revised manuscript to you.

Comment: Overall, I understand that you categorize climate into distinct categories (classes) for analytical reasons, but these are still no real physical entities with clear boundaries. This still needs to be highlighted more clearly at the onset of the methods (§ data collection).

Response: We have clarified this by adding the requested text to the manuscript.

Comment: Ensure that figures are intelligible (also supplementary). This typically means increasing font sizes, increasing line widths, and adapt color schemes in R to something a bit more color-blind friendly, or at least with more contrast between adjacent colors. There were so many different hues of green, blue, reddish, pink in some supplemental figures that they became hard to understand (suppl figs 1 and 3 in particular). Also, please explain abbreviations in supplemental figures too.

Response: We have remade figure 2 and all the supplemental figures to be more understandable and legible.